# Bloodstream Infection Caused by *Erysipelothrix rhusiopathiae* in an Immunocompetent Patient

**DOI:** 10.3390/microorganisms12050942

**Published:** 2024-05-07

**Authors:** Irene Mileto, Cristina Merla, Marta Corbella, Stefano Gaiarsa, Angela Kuka, Stefania Ghilotti, Pasquale De Cata, Fausto Baldanti, Patrizia Cambieri

**Affiliations:** 1Department of Microbiology & Virology, Fondazione IRCCS Policlinico San Matteo, 27100 Pavia, Italy; 2Specialization School of Microbiology and Virology, University of Pavia, 27100 Pavia, Italy; 3Department of General Medicine, Istituti Clinici Maugeri IRCCS, 27100 Pavia, Italy; 4Department of Clinical, Surgical, Diagnostic and Pediatric Sciences, University of Pavia, 27100 Pavia, Italy

**Keywords:** *Erysipelothrix rhusiopathiae*, bacteraemia, bloodstream infection

## Abstract

*Erysipelothrix rhusiopathiae* is a facultative anaerobe Gram-positive bacillus, which is considered a zoonotic pathogen. *E. rhusiopathiae* causes erysipeloid, mainly in occupational groups such as veterinarians, slaughterhouse workers, farmers, and fishermen. Two cutaneous forms (localised and generalised) and a septicaemic form have been described. Here, we report the isolation of a strain of *E. rhusiopathiae* from a 56-year-old immunocompetent obese male admitted to Fondazione IRCCS Policlinico San Matteo Pavia (Italy). Blood cultures were collected and Gram-positive bacilli were observed. *E. rhusiopathiae* grew and was identified. Antimicrobial susceptibility tests were performed and interpreted with EUCAST breakpoints (PK-PD). The strain was susceptible to all the antibiotics tested, while it was intrinsically resistant to vancomycin. The clinical diagnosis of *E. rhusiopathiae* can be challenging, due to the broad spectrum of symptoms and potential side effects, including serious systemic infections such as heart diseases. In the case described, bacteraemia caused by *E. rhusiopathiae* was detected in a immunocompetent patient. Bacteraemia caused by *E. rhusiopathiae* is rare in immunocompetent people and blood cultures were proven to be essential for the diagnosis and underdiagnosis of this pathogen, which is possible due to its resemblance to other clinical manifestations.

## 1. Introduction

*Erysipelothrix* spp. are rod-shaped Gram-positive facultative anaerobic bacteria. Only eight species belong to this genus [1]. Among them, *Erysipelothrix rhusiopathiae* is the most frequently isolated species. *E. rhusiopathiae* is primarily known as a zoonotic pathogen. Swine serves as the main reservoir, but it can also be isolated from domestic animals, fish, and birds [2,3]. *E. rhusiopathiae* can also infect mammals, reptiles, and insects. Human infections are associated with exposure to animals and usually infect individuals belonging to certain occupational groups, such as veterinarians, butchers, farmers, and fishermen [4,5,6,7]. Erysipeloid typically manifests on the face or legs as a raised, well-demarcated, bright red rash [8]. Three forms of erysipeloid have been described, as follows: the localised and the generalised forms, which are cutaneous, and the septicaemic form, [9] associated with endocarditis in 90% of cases [9,10]. However, not all the bacteraemia caused by *E. rhusiopathiae* develop into endocarditis [11]. Although *E. rhusiopathiae* infections are not common in humans, Rostamian and colleagues reported an increasing number of isolations in recent years [12]. Here, we report the isolation of a strain of *Erysipelothrix rhusiopathiae* from a blood sample taken from a 56-year-old male admitted to Fondazione IRCCS Policlinico San Matteo Pavia (Italy) with a cutaneous rash.

## 2. Case Description

A 56-year-old obese male was admitted to the Emergency Room (ER) after an accidental fall. The patient presented with a cutaneous rush on his lower limbs and abdomen and had comorbidities such as dyslipidaemia, hypertension, and hepatic steatosis. The patient had no head injury, but he lost consciousness and reported asthenia. He was febrile, so blood cultures were collected immediately. Two sets of blood cultures were incubated in the BD BACTEC FX automated blood culture system (Becton Dickinson and Company, Franklin Lakes, NJ, USA), according to the manufacturer’s instructions. Urine was also cultured. High values of C-Reactive protein (14.46 mg/dL) and procalcitonin (1.27 mg/dL) were measured and a chest X-ray was performed. Piperacillin/tazobactam and clindamycin were started in the ER. On the same day, the patient was transferred to the Istituti Clinici Scientifici Maugeri (Pavia, Italy).

On day 2, an aerobic blood culture bottle showed microbial growth after 18 h and 51 min of incubation. A Gram stain was performed and Gram-variable bacilli were observed (Figure 1). The positive blood culture was streaked on Columbia agar +5% sheep blood, Chocolate agar + PolyViteX™, and Schaedler agar + 5% sheep blood (bioMérieux SA, Marcy-l’Etoile, France). The plates were incubated overnight at 36 ± 1 °C, according to laboratory procedures. On the same day, oral metronidazole was administered after a consultation with a dermatologist. The specialist also recommended daily antibiotic bandages containing gentamicin and zinc oxide, as well as the application of Vaseline oil on the lesions.

On day 3, *Erysipelotrix rhusiopathiae* grew from the culture (Figure 1) and it was identified using Matrix-Assisted Laser Desorption Ionisation Time-Of-Flight (MALDI-TOF) (Bruker Daltonics GmbH, Bremen, Germany), equipped with BioTyper version 3.0. Antimicrobial susceptibility was tested on the strain of *E. rhusiopathiae* (PV7573) using the E-test concentration gradient diffusion assay on Mueller–Hinton Fastidious (Liofilchem, Roseto degli Abruzzi, Italy) and was incubated overnight at 37 °C, starting from a bacterial suspension with a turbidity of 0.5 McFarland, as indicated by the European Committee on Antimicrobial Susceptibility Testing, EUCAST, Version 12.0, 2022 [13]. Minimal Inhibitory Concentration values were interpreted according to non-species-related breakpoints, based on pharmacokinetics/pharmacodynamics (PK/PD section) (EUCAST, Version 12.0, 2022) [13]. The strain, named PV7573, was susceptible to benzylpenicillin, piperacillin/tazobactam, ceftriaxone, imipenem, and ciprofloxacin (Table 1). For metronidazole, tetracycline, vancomycin, clindamycin, and erythromycin, no breakpoints were available.

On day 4, when the antibiotic susceptibility profile was available, metronidazole and clindamycin were stopped, according to the consultation with the infectious disease specialist. Piperacillin/tazobactam was administered for a further thirteen days and clinical improvement was progressive. The patient was discharged on day 20, in a good health condition.

In addition to the clinical procedures, we sequenced the whole genome of the isolate with the Illumina platform, to complete the characterisation of the case. Illumina short reads were assembled into a 1,789,669 bp draft genome (strain name PV7573) using Shovill 1.1 (https://github.com/tseemann/shovill; accessed on 16 July 2023). Virulence genes were manually checked by searching in the genome of PV7573 for the sequences of *spaA*, *spaB*, *cpsABC*, *hylA*, *nanH.1*, *nanH.2*, and *rspAB* (all from Erysipelotrix rhusiopathiae str. Fujisawa; NC_015601.1). PV7573 showed *spaA* with an identity of 98%, as well as *hylA*, *nanH.1*, *nanH.2*, *cpsA*, and *rspAB* with an identity of 99%. No antimicrobial resistance gene was detected using either Resfinder [14] (https://cge.food.dtu.dk/services/ResFinder/, accessed on 7 September 2023) or AMRFinderPlus [15] (https://www.ncbi.nlm.nih.gov/pathogens/antimicrobial-resistance/AMRFinder/, accessed on 7 September 2023).

The genome is available in the NCBI database under Bioproject PRJNA1055344. The serotype was determined through the online tool In silico PCR amplification [16] using the serotype-specific primers previously described by Shiraiwa and colleagues [17] and Shimoji and colleagues [18]. PV7573 belonged to serotype 9.

Looking for epidemiological insights, we performed the core-SNPs-based maximum likelihood phylogeny inference of the novel isolate and all 20 high-quality *E. rhusiopathiae* genomes available in the BV-BRC database (last accessed on 17 September 2023), using the P-DOR pipeline (Figure 2) [19]. Strain PV7573 is most closely related to strain 10DISL, isolated in the USA in 2020 from wildlife.

## 3. Discussion and Conclusions

In this study, we report a case of bacteraemia caused by *E. rhusiopathiae* in an immunocompetent patient, who was successfully treated with piperacillin tazobactam.

Infections caused by *E. rhusiopathiae* are uncommon, as reported in a systematic review in 2021 by Rostamian and colleagues. In the literature, they found only 62 cases (75.8% male and 24.2% female), with an average age of 54.16 years [12]. *E. rhusiopathiae* usually causes infections in people with animal-related occupations [4,5,17], as also observed by Rostamian and colleagues. Although they found that the occupation of 48.4% of cases was not clearly identified, they reported that, among the cases with known occupation, 14.5% were farmers, 9.7% had fish/seafood-related jobs, and 3.2% were butchers. However, 59.7% of the cases reported a history of contact with animals [12]. Although animal contact is the main transmission route [20,21], this bacterium can persist for several months in many environmental sources such as soil and water, as well as in decaying animals [22,23]. In our case, the origin of the pathogen was not identified, but our patient did not report any animal contact and lived in non-optimal hygienic conditions. This situation was also observed in other cases, as described by other authors [11,12].

Bacteriaemia due to *E. rhusiopathiae* is uncommon and it usually occurs in immunocompromised patients, especially those who suffer from chronic kidney disease, diabetes mellitus, and/or those submitted to treatment with immunosuppressive drugs [5]. The patient in this study was immunocompetent, but he had comorbidities such as obesity, dyslipidaemia, hypertension, and hepatic steatosis. He was admitted to the ER with generic symptoms such as fever, hypertension, and a peculiar cutaneous rash on the lower limbs and abdomen. Although the patient in this study was immunocompetent, the strain was able to enter the bloodstream. For this reason, the whole genome of the strain was sequenced to investigate the presence of specific virulence and resistance genes. PV7573 displayed the presence of several genes encoding for virulence factors such as surface protection antigen A (spaA), which acts as an adhesin in *E. rhusiopathiae*; the glycosyltransferase; capsule polysaccharide synthesis (cpsA-C); *E. rhusiopathiae* surface protein (rspA and rspB); hyaluronidase (hylA-C); and neuraminidase (nanH.1 and nanH.2). However, most of the genomes available on the BV-BRC database show the same virulence factors, indicating that the strain isolated in this study was not particularly virulent. No antimicrobial resistance genes were observed through the genomic analyses. However, there is a scarce knowledge of resistance determinants for this species, because few strains of *E. rhusiopathiae* have been characterised from both phenotypic and genomic sides, so far.

Regarding the antimicrobial resistance profile, PV7573 was susceptible to all the antibiotics for which non-species-related PK/PD EUCAST breakpoints were available. As others studies have reported [16,17], penicillin has been reported to be very efficient in treating *E. rhusiopathiae* infections, even at low dosages. Moreover, Brooke and Riley report that oral penicillin can resolve erysipeloid in 48 h [20]. Ceftriaxone, imipenem, fluoroquinolone, and clindamycin are also considered effective antimicrobials [24,25]. The strain PV7573 resulted in being susceptible to penicillin, ceftriaxone, imipenem, and ciprofloxacin; no interpretation was available for clindamycin, even if the MIC value for this antibiotic was very low (0.064 µg/mL). Clindamycin and erythromycin are only bacteriostatic for *E. rhusiopathiae* [4]. However, piperacillin/tazobactam, which was started as empiric therapy upon admission, was continued until the end of the hospitalisation, since it was proven to be effective.

High values of MIC were measured for vancomycin and metronidazole. According to the literature, *E. rhusiopathiae* is intrinsically resistant to vancomycin, a broad-spectrum drug frequently used for the treatment of infections caused by Gram-positive bacteria [23]. Metronidazole was used in this case as an empiric therapy, with piperacillin/tazobactam and clindamycin, until the antimicrobial susceptibility profile was available.

In conclusion, the clinical diagnosis of *E. rhusiopathiae* could be challenging due to the wide spectrum of manifestations and the possible complications such as cardiac diseases, especially endocarditis [5]. In the case described, the patient did not present involvement of the endocardium. Blood cultures were proven to be crucial for the diagnosis of *E. rhusiopathiae* infection, considering the unusual clinical manifestations. Moreover, *E. rhusiopathiae* bloodstream infection may occur more commonly than it is reported in the literature. Indeed, Gram-positive rods cultured from blood can be considered as contamination by *Corynebacterium* spp, leading to the underdiagnosis of this bacterial infection. Additionally, another possible cause of underdiagnosis of infections as a result of *E. rhusiopathiae* is the resemblance to erysipelas caused by *Staphylococcus aureus* and beta haemolytic streptococcal infections [11,26]. With *E. rhusiopathiae* being resistant to vancomycin, it is important to correctly identify it, even more so in cases of endocarditis, for which this antibiotic is often used as an empirical therapy.

Finally, for the future, it is desirable to complement routine diagnosis methods with the whole genome sequencing of *Erysipelothrix* strains, to explore the characteristics of this bacterial species.

## Figures and Tables

**Figure 1 microorganisms-12-00942-f001:**
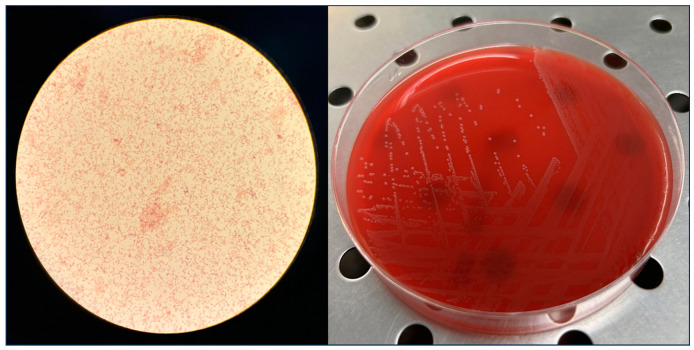
Gram staining of *E. rhusiopathiae* and morphology of its colonies on Columbia agar +5% sheep blood, after incubation at 37 °C for 24 h.

**Figure 2 microorganisms-12-00942-f002:**
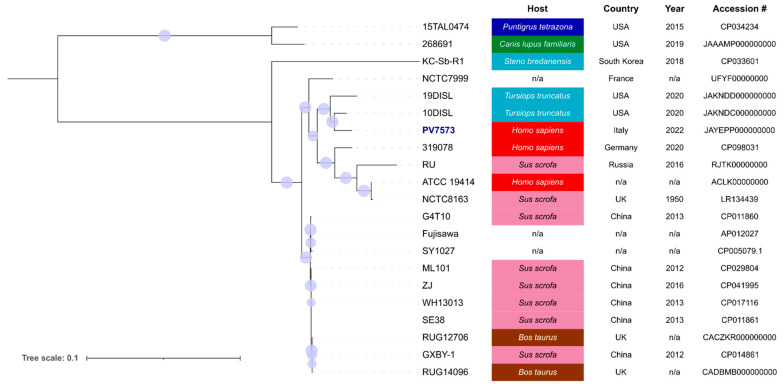
Phylogenetic analysis obtained with P-DOR pipeline of the PV7573 isolate and all 20 high-quality *E. rhusiopathiae* genomes available in the BV-BRC database (last accessed on 17 September 2023). Source, year, and country of isolation are reported alongside the nodes. Bootstrap values above 70 are represented with dots on the tree branches.

**Table 1 microorganisms-12-00942-t001:** Antibiotic susceptibility profile of the strain PV7573. Interpretation was performed according to EUCAST breakpoints (version 12, 2022). Abbreviations were used for the following antibiotics: benzylpenicillin (PNG); piperacillin/tazobactam (TZP); ceftriaxone (CRO); imipenem (IMI); ciprofloxacin (CIP); metronidazole (MTZ); tetracycline (TET); vancomycin (VAN); clindamycin (CC); erythromycin (EE). S was used for susceptible, while “-” was used when non-species-related PK/PD breakpoints were not available. PK/PD breakpoints are shown under the abbreviation of each antibiotic, when available.

Strain	PNG (R > 2)	TZP (R > 16)	CRO (R > 2)	IMI (R > 4)	CIP (R > 0.5)	MTZ	TET	VAN	CC	EE	D Test
**PV7573**	0.125 (S)	0.064 (S)	0.064 (S)	0.006 (S)	0.032 (S)	>256 (-)	0.5 (-)	32 (-)	0.064 (-)	0.025 (-)	Negative

## Data Availability

The genome is available from the NCBI database under Bioproject PRJNA1055344.

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
