# Peer review of "Bloodstream Infection Caused by Erysipelothrix rhusiopathiae in an Immunocompetent Patient"

_microorganisms, 2024, doi:10.3390/microorganisms12050942_

Round 1

Reviewer 1 Report

Comments and Suggestions for Authors

Dear authors,

Very interesting manuscript.

The case was presented in a clear way. Aczkowliek I have some comments that result from various perspectives ("fields") of medicine we deal with.

From my perspective as a veterinarian and microbiology: yes, E. Rhusiopathiae is g+ but in direct preprats from clinical material or even more microbiological breeding it cannot be mistaken for Corynebacterium. What I have durable myself many times by isolating this bacterium, Erysipelothrix in the preparation is a gram variable, these are long, threads.

That is why I always encourage you to add photos (in a supplement) from a potential change in the skin (if it is possible) photos from microbiological cultures and stained preparations.

I cannot agree that erysipelas and erysipeloid lesions can be confused with each other.

In cases where we have a typical animal pathogen, the zoonotic nature of the pathogen should be emphasized and the need to find the source of the pathogen should be indicated.

In veterinary medicine, the standard antibiotic used for diamond skin disease is penicillin. Where a single dose is often enough.

Author Response

Reviewer 1

Dear authors,

Very interesting manuscript.

The case was presented in a clear way. Aczkowliek I have some comments that result from various perspectives ("fields") of medicine we deal with.

From my perspective as a veterinarian and microbiology: yes, E. Rhusiopathiae is g+ but in direct preprats from clinical material or even more microbiological breeding it cannot be mistaken for Corynebacterium. What I have durable myself many times by isolating this bacterium, Erysipelothrix in the preparation is a gram variable, these are long, threads. That is why I always encourage you to add photos (in a supplement) from a potential change in the skin (if it is possible) photos from microbiological cultures and stained preparations.

Thank you for your observation. We wrote Gram positive since the bacterium is recognized in literature as a Gram positive. However, we added a picture of the Gram staining and of the cultures. We agree that Erysipelothrix spp. cannot be mistaken for Corynebacterium, that we see very frequently, due to the peculiar arrangement of bacilli in Corynebacterium spp.

 I cannot agree that erysipelas and erysipeloid lesions can be confused with each other.

This is the first isolation of Erysipelothrix spp. in our hospital which had 1000 beds and it serves all the Pavia province (number inhabitants: 545888 in 2019, data from Eurostat). The isolation of this bacterial species is rare in our area and clinicians are not used to discriminate between erysipelas and erysipeloids. We added some reference to our statement.

In cases where we have a typical animal pathogen, the zoonotic nature of the pathogen should be emphasized and the need to find the source of the pathogen should be indicated.

Thank you for the observation. We described the zoonotic nature of the pathogen in the introduction. However, we have chosen to not emphasize that since the patient did not report a direct contact with animals. We decided to state clearly in the text at line 117-119 that the patient did not have contact with animals, but he lived in non-optimal hygienic conditions.

In veterinary medicine, the standard antibiotic used for diamond skin disease is penicillin. Where a single dose is often enough.

The patient presented to the ER with fever and a cutaneous rush, and it was not possible to know at that time that the infection was caused by Erysipelothrix rhusiopathiae. There, he received an empiric broad-spectrum treatment with piperacillin/tazobactam and clindamycin and blood cultures were collected (day 1). Due to the lesion the dermatologist who visited the patient added metronidazole and bandages containing gentamycin for the lesion. Once E. rhusiopathiae grown on the plate (day 3) identification and antibiogram were performed. The result of the antibiogram was available on day 4 and only piperacillin/tazobactam which was proven to be susceptible was continued. Probably if there was only the presence of the lesion a targeted therapy with penicillin or cephalosporins would have been started after the identification of the pathogen.

Reviewer 2 Report

Comments and Suggestions for Authors

The manuscript contains a case report of diagnosis and treatment of a patient infected with Erysipelothrix rhusiopathiae. The case is quite interesting because blood infections in humans caused by these bacteria are rarely reported. However, the way the manuscript was written and the research methodology used leave much to be desired.

The authors did not specify what method they used to determine the sensitivity of bacteria to antimicrobial substances - MIC values can be determined using various methods, and the recommended method for E. rhusiopathie is the broth dilution method (CLSI Vet06).

What were the authors' considerations when selecting antimicrobials for AST?

Secondly, the EUCAST guide referred to by the authors does not contain recommendations (cut-off points) for bacteria of the Erysipelothrix genus. It is also not true that there are no established breakpoints for erythromycin and clindamycin (L74), because the cutoff points for these antibiotics (as well as for several others) are included in the CLSI VEt06 guidelines for E. rhusiopathiae.

Why did the authors not determine the serotype of the strain tested?

L94 - what was the phylogenetic analysis based on? For whole genome analysis, the cortical genome is typically analyzed.

Figure 1. - It is not known what the digits/numbers mean; in the text, the authors write about the 10DISL strain (L95), the name of which is not shown in the dendrogram. The dendrogram should include the names of the strains, accession numbers, country of isolation and host species and possibly serotype; the manuscript must also contain accession numbers of the analyzed genomes - they can be provided on the dendrogram or in the supplement.

What exactly was the purpose of whole genome sequencing?

L120 - the statement that the PV7573 strain "displayed only the most common virulence factors" is quite unclear; Did the authors detect virulence factors produced by this strain or did they only analyze the presence of genes encoding these virulence factors in the genome? How many and which virulence genes were detected? Which were absent in the geome of the strain studied?  In-depth analyzes of virulence factors are indicated due to the atypical nature of the infection.

What method was used to detect resistance genes?

L135 - is vancomycin really used as a first-line drug in Italy?

L65 - the use of gentamicin in the case of E. rhusiopathiae infection (or even suspected infection) is reprehensible, because, as many researchers have proven, this bacterium is naturally resistant to gentamicin. This method of treatment should not be promoted.

L148 - streptococcal - with a lowercase letter

What are the statistics of E. rhusiopathiae infections in humans?

Author Response

Reviewer 2

The manuscript contains a case report of diagnosis and treatment of a patient infected with Erysipelothrix rhusiopathiae. The case is quite interesting because blood infections in humans caused by these bacteria are rarely reported. However, the way the manuscript was written, and the research methodology used leave much to be desired.

The authors did not specify what method they used to determine the sensitivity of bacteria to antimicrobial substances - MIC values can be determined using various methods, and the recommended method for E. rhusiopathie is the broth dilution method (CLSI Vet06).

Thank you for the comment. We added in the text the method that we used concentration gradient diffusion assay. The case described in this study happened in Italy, where the official guidelines for antimicrobial suspeptibility interpretation is by EUCAST. The guidelines of CLSI Vet06 are for microorganisms isolated from animals and we do not use them for patients.

What were the authors' considerations when selecting antimicrobials for AST?

We selected antimicrobials generally used for the treatment of Gram-positive bacilli. Moreover, we searched in the literature the antimicrobial treatment suitable for E. rhusiopathiae infections, and we also considered the empiric therapy that the patient was receiving.

Secondly, the EUCAST guide referred to by the authors does not contain recommendations (cut-off points) for bacteria of the Erysipelothrix genus. It is also not true that there are no established breakpoints for erythromycin and clindamycin (L74), because the cutoff points for these antibiotics (as well as for several others) are included in the CLSI VEt06 guidelines for E. rhusiopathiae.

Since for us  living in Italy EUCAST guidelines are the reference guidelines in Europe, we decided to interpretate the susceptibility with PK/PD breakpoints which was still possible at that time. We agree that the breakpoints exist in CLSI Vet06, but they were not used in this case since that breakpoints were for bacteria isolated from animals.

Why did the authors not determine the serotype of the strain tested?

We determined the serotype from the whole genome.

L94 - what was the phylogenetic analysis based on? For whole genome analysis, the cortical genome is typically analyzed.

The phylogeny was inferred using a coreSNP alignment. We amended the text to clarify this.

Figure 1. - It is not known what the digits/numbers mean; in the text, the authors write about the 10DISL strain (L95), the name of which is not shown in the dendrogram. The dendrogram should include the names of the strains, accession numbers, country of isolation and host species and possibly serotype; the manuscript must also contain accession numbers of the analyzed genomes - they can be provided on the dendrogram or in the supplement.

Numbers on tree branches are bootstrap values. We amended the figure legend and edited the figure to make it clearer and more informative, as requested.

What exactly was the purpose of whole genome sequencing?

This is the first isolation of Erysipelothrix spp. in our hospital which had 1000 beds and it serves all the Pavia province (number inhabitants: 545888 in 2019, data from Eurostat). Due to the infrequent isolation of this species we decided to perform the sequencing.

L120 - the statement that the PV7573 strain "displayed only the most common virulence factors" is quite unclear; Did the authors detect virulence factors produced by this strain or did they only analyze the presence of genes encoding these virulence factors in the genome? How many and which virulence genes were detected? Which were absent in the geome of the strain studied?  In-depth analyzes of virulence factors are indicated due to the atypical nature of the infection.

Thank you for the observation, we changed the sentence in the text. We detected the genes encoding for virulence factor and we added this information at line 105-109. You are correct. The presence of virulence genes was investigated due to the atypical nature of the infection.

What method was used to detect resistance genes?

Added in the text.

L135 - is vancomycin really used as a first-line drug in Italy?

Thank you for your correction. We modified the text.

L65 - the use of gentamicin in the case of E. rhusiopathiae infection (or even suspected infection) is reprehensible, because, as many researchers have proven, this bacterium is naturally resistant to gentamicin. This method of treatment should not be promoted.

Thank you for your observation. As written at line 66, gentamycin was started on day 2, while the identification of E. rhusiopathiae from blood cultures happened on day 3. Bandages containing gentamycin were stopped on that day.

L148 - streptococcal - with a lowercase letter

Thank you. Corrected.

What are the statistics of E. rhusiopathiae infections in humans?

No statistics are available. We cited the data about a systematic review in the text at line 133.

Reviewer 3 Report

Comments and Suggestions for Authors

The manuscript by Irene Mileto et al. entitled "Bloodstream infection caused by Erysipelothrix rhusiopathiae in an immunocompetent patient" is a case report devoted to the characterization of the Erysipelothrix rhusiopathiae strain isolated from the patient and sequenced by WGS. The manuscript is not without flaws.

Although the manuscript mentions whole genome sequencing of strains, it does not provide details on how this was done, how the DNA was isolated, what specific apparatus was used, how assembly was performed, analysis of virulence factors, phylogenetic analysis, etc.

Other issues:

Line 42: Why is Rostamian in italics?

Line 44: 1.27

Line 99: How was the phylogenetic analysis done, using which genes, which method? Why is there no outgroup? What is the P-DOR pipeline?

Line 103: Erysipelothrix abbreviated to E.

Comments on the Quality of English Language

Since English is not the native language of any of the authors, careful English editing is recommended.

Author Response

Reviewer 3

The manuscript by Irene Mileto et al. entitled "Bloodstream infection caused by Erysipelothrix rhusiopathiae in an immunocompetent patient" is a case report devoted to the characterization of the Erysipelothrix rhusiopathiae strain isolated from the patient and sequenced by WGS. The manuscript is not without flaws.

Although the manuscript mentions whole genome sequencing of strains, it does not provide details on how this was done, how the DNA was isolated, what specific apparatus was used, how assembly was performed, analysis of virulence factors, phylogenetic analysis, etc.

Thanks, we added in the methods.

Other issues:

Line 42: Why is Rostamian in italics?

Corrected in the text.

Line 44: 1.27

Corrected in the text.

Line 99: How was the phylogenetic analysis done, using which genes, which method? Why is there no outgroup? What is the P-DOR pipeline?

The phylogeny was inferred using IQtree (Maximum Likelihood) on a coreSNP alignment. The entire process is performed within the P-DOR pipeline. We have expanded the maintext and fixed the unreferenced citation.

We have decided not to use an outgroup to avoid the loss of too many core position in the alignment. Our interest was not to reconstruct the evolutionary history of the entire species; we focused on our novel isolate/genome, so we privileged local resolution in the tree topology at the expense of global clarity. The tree was rooted at midpoint

Line 103: Erysipelothrix abbreviated to E.

Done, thank you.

Comments on the Quality of English Language

Since English is not the native language of any of the authors, careful English editing is recommended.

Done

Round 2

Reviewer 2 Report

Comments and Suggestions for Authors

See attachment

Author Response

 The Authors improved the manuscript and performed additional analyses, but several elements still require correction.

L276 – do you mean Etest? If yes add „(Etest)” after concentration gradient diffusion assay, please;

Done

L276 – What microbiological medium was the "concentration gradient diffusion assay" performed on? What was the density of the bacterial suspension? How long were the bacteria incubated? (How long did it take for the result to be read?)

Added

L277 - to increase clarity, add information that the interpretation of AST was made according to non-species related breakpoints based on pharmacokinetics/pharmacodynamics (PK/PD section) (European Committee on Antimicrobial Susceptibility Testing, EUCAST, Version 12.0, 279 2022).

Modified.

L277 – add numer of ciatation (for EUCAST guedline)

Reference section should include the cytacjÄ™ dokumentu stosowanego do interpretacji wyników lekowrażliwoÅ›ci, tj. European Committee on Antimicrobial Susceptibility Testing, EUCAST, Version 12.0, 279 2022

Added.

In the first row of Table 1, under the symbols of antimicrobial substances, I suggest adding cut-off values indicating resistance, so that the reader can see at a glance the difference between the cut-off and the obtained MIC value, e.g.

PNG

R>2

Done.